# Stable Basis Deep Neural Policy Training

## Abstract

Following the initial success of deep reinforcement learning in learning policies just by interacting with complex, high-dimensional representations, and a decade of significant research, deep neural policies have been applied to a striking variety of fields ranging from pharmaceuticals to foundation models that can reason. Nonetheless one of the strongest assumptions of reinforcement learning is that a reward signal will be available in the MDP. While this assumption comes in handy in certain fields, such as automated financial markets, it does not naturally fit in many others where the computational complexity of providing such a signal for the task at hand is greater than the complexity of learning one. In this paper we focus on learning policies in MDPs without this assumption, and study sequential decision making without having access to information on rewards provided by the MDP. We introduce a training method in high-dimensional MDPs and provide a theoretically well-founded algorithm that significantly improves the sample complexity of deep neural policies. The theoretical and empirical analysis reported in our paper demonstrates that our method achieves substantial improvements in sample efficient training while constructing more stable and resilient policies that can generalize to uncertain environments.

## 1 Introduction

Interacting with a given environment solely based on observations and receiving rewards upon taking actions in high-dimensional state observation MDPs gained substantial acceleration with the recent advancements in deep reinforcement learning research (Mnih et al., 2016; Kapturowski et al., 2023; Abel et al., 2023; Flennerhag et al., 2023). Currently, from automated financial markets to solving complex games (Schrittwieser et al., 2020) to designing algorithms (Fawzi et al., 2022; Mankowitz et al., 2023), a broad range of fields varied from pharmaceuticals (Popova et al., 2018; Korshunova et al., 2022) to self-operating vehicles and large language models (Touvron et al., 2023; Pang et al., 2024; Su et al., 2025) have benefited from the advancements achieved in deep sequential decision making algorithms that can learn functioning policies in high-dimensional observation MDPs. Despite recent progress, a fundamental assumption persists in reinforcement learning: that the agent has direct access to the reward function of the MDP. From bee foraging to human decision making the reward signal of natural intelligence is complicated and a non-stationary function of manifold inputs (Doya & Sejnowski, 1994; Montague et al., 1995; Schmajuk & Zanutto, 1997). Towards targeting to surpass the skills and capabilities of natural intelligence, now we are in the era of probing these fundamental assumptions.

Analyzing the amount of experiences that needs to be obtained by the policy to have the skills necessary to function effectively in a specific environment is one of the foundational questions that has been studied extensively (Fiechter, 1994; Kearns & Singh, 1999; Kakade, 2003). Recent studies argued that policies trained in the absence of a reward signal can in fact learn faster. Orthogonal to these advances while the instabilities of deep neural networks under non-robust directions have been a subject of discussion (Goodfellow et al., 2015), recent work demonstrated that these instabilities are currently also present in deep neural policies (Huang et al., 2017). Furthermore, more recent studies demonstrated that these non-robust directions can be semantically meaningful changes to the environment (Korkmaz, 2024). Thus, in this paper we focus on the sample-efficiency and robustness of the policies that can learn functioning strategies without the reward signal provided by the MDP and ask the following questions:

- *How can we build agents that can learn neural policies with fewer interactions?*

- *What are the foundational building blocks towards constructing policies that can make resilient and robust decisions in unstable and non-robust environments?*
- *How can we analyze and quantify the robustness of deep sequential decision making policies with high-dimensional state observations spectrally?*

Hence, to answer these questions in this paper we focus on environment interactions of deep neural policies, and make the following contributions.

**Contributions.** In this paper, we introduce harmonic learning, a theoretically well-founded algorithm grounded in stable basis training that substantially improves the sample complexity of deep sequential decision making algorithms and learns policies that are stable and robust. We first provide the theoretical analysis in Section 3, and explain why and how stable basis training improves sample-efficiency. We then conduct experiments in the Arcade Learning Environment (ALE) with high dimensional state observation MDPs in Section 4, and the experimental results reported in our paper verify the theoretical analysis. Harmonic learning achieves substantial sample-efficiency resulting in requiring up to $20\times$ fewer samples while achieving better performance. We further introduce a novel method, grounded in theory, to spectrally analyze the robustness of deep neural policies. We contrast and compare policy volatilities and instabilities of the state-of-the-art imitation learning to standard reinforcement learning in high dimensional state representation MDPs. Our method reveals the spectral contrast between the vanilla deep reinforcement learning policies and the state-of-the-art deep sequential decision making policies that can learn without a reward function. Finally, we analyze the generalization capabilities, natural robustness and overfitting of the state-action value function and our analysis further demonstrates that harmonic learning leads to policies that are substantially more robust and generalizable.

## 2 BACKGROUND AND PRELIMINARIES

**Preliminaries.** A Markov Decision Process (MDP) is represented as a tuple $\langle \mathcal{S}, \mathcal{A}, \mathcal{P}, r, \gamma, \tau_0 \rangle$ of a set of states $\mathcal{S}$, a set of actions $\mathcal{A}$, transition probability distribution $\mathcal{P}(s_{t+1}|s_t, a_t)$, and a reward function $r : \mathcal{S} \times \mathcal{A} \to \mathbb{R}$, discount factor $\gamma$, and initial state distribution $\tau_0$. The objective in reinforcement learning is to learn a policy that will maximize the expected discounted cumulative rewards obtained by the policy $\pi : \mathcal{S} \to \mathcal{P}(\mathcal{A})$. This objective can be achieved via $\mathcal{Q}$-learning that essentially learns a $\mathcal{Q}$ function $\mathcal{Q} : \mathcal{S} \times \mathcal{A} \to \mathbb{R}$ that will assign values to each state-action $(s, a)$ pair to reveal what would be the expected cumulative discounted rewards obtained if the action $a$ is taken in state $s$. The $\mathcal{Q}$-function is learnt via iterative Bellman update $\mathcal{Q}(s_t, a_t) = r(s_t, a_t, s_{t+1}) + \gamma \sum_{s_t} \mathcal{P}(s_{t+1}|s_t, a_t) \max_a \mathcal{Q}(s_{t+1}, a)$ (Watkins, 1989). The value function is defined to be $\mathcal{V}(s) = \max_a \mathcal{Q}(s, a)$. Upon the construction of the state-action value function the policy executes the action that maximizes the state-action value function $\hat{a} = \arg\max_{a \in A} \mathcal{Q}(s, a)$. In settings where the state or action space have high-dimensional representations, the state-action value function is approximated via a deep neural network.

$$\theta_{t+1} = \theta_t + \alpha(r(s_t, a_t, s_{t+1}) + \gamma \max_a \mathcal{Q}(s_{t+1}, a; \theta_{t+1}) - \mathcal{Q}(s_t, a; \theta_t))\nabla_{\theta_t} \mathcal{Q}(s_t, a; \theta_t)$$

For a given setting where the reward function is not present, the reward function can be estimated from observing trajectories of a functioning policy, i.e. inverse reinforcement learning. The first study that proposed this concept achieves this objective via linear programming (Ng & Russell, 2000).

$$\max \sum_{s \in \mathcal{S}_\rho} \min_{a \in \mathcal{A}} \{\Delta(\mathbb{E}_{s' \sim \mathcal{P}(\cdot|s,a_1)} \mathcal{V}^\pi(s') - \mathbb{E}_{s' \sim \mathcal{P}(\cdot|s,a)} \mathcal{V}^\pi(s'))\}$$

subject to $|\alpha_i| \leq 1$, $i = 1, 2, \ldots, d$, where $\Delta(x) = x$ if $x > 0$ and $\Delta(x) = 2x$ otherwise. While some studies focused on learning the reward function itself others focused on directly learning a policy from demonstrations (Kostrikov et al., 2020). Quite recently, Garg et al. (2021) focused on learning a state-action value function via solely observing the trajectories of a functioning policy (inverse $\mathcal{Q}$-learning), and maximizing the objective function $\mathcal{J}(\theta)$ given by

$$\mathbb{E}_{(s,a) \sim \rho_E} \left[ \phi \left( \mathcal{Q}_\theta(s, a) - \gamma \mathbb{E}_{s' \sim \mathcal{P}(\cdot|s,a)}[\mathcal{V}_\theta(s')] \right) \right] - \mathbb{E}_{(s,a) \sim \mu} \left[ \phi \left( \mathcal{V}_\theta(s) - \gamma \mathbb{E}_{s' \sim \mathcal{P}(\cdot|s,a)}[\mathcal{V}_\theta(s')] \right) \right]$$

where $\rho_E$ is the occupancy measure of the expert policy, and $\mu$ is any valid occupancy measure. The method introduced in this paper achieves state-of-the-art performance in environments with high-dimensional observations. Moreover, the authors of this study argue that once the state-action value

function, i.e. $\mathcal{Q}(s,a)$, is learnt, the reward function, i.e. $r(s_t, a_t, s_{t+1})$, can be reconstructed from this information. Furthermore, note that the inverse $\mathcal{Q}$-learning algorithm can learn a functioning policy and a reward function simultaneously; hence, throughout the paper the inverse $\mathcal{Q}$-learning algorithm will be referred to as an imitation and inverse reinforcement learning algorithm interchangeably.

**Robustness in Deep Reinforcement Learning.** Vulnerabilities and robustness of deep reinforcement learning policies were initially discussed in Huang et al. (2017). This study essentially introduces fast gradient sign method produced adversarial perturbations (Goodfellow et al., 2015) into the observations of the deep reinforcement learning policies. In this line of research some studies tried to further identify adversarial directions (Korkmaz & Brown-Cohen, 2023), while others focused on solving the robustness problem via training with these adversarial directions (Gleave et al., 2020; Pinto et al., 2017). Recently, some studies demonstrated that the adversarial directions are shared across states, across MDPs and across algorithms (Korkmaz, 2022). Moreover, the adversarially trained deep reinforcement learning policies inherit the exact same adversarial directions with the vanilla trained deep reinforcement learning policies. While there are some studies working on the diagnostic perspective of robustness in deep reinforcement learning by using the Carlini & Wagner (2017) formulation, these studies highlight that certified adversarial training shifts vulnerabilities towards a different band in the frequency spectrum instead of eliminating these non-robust features (Korkmaz, 2024). In connection to this, some studies focused on demonstrating the contrast between adversarial and natural directions in terms of their perceptual similarities to the base state observations and the impact they can cause on the policy performance (Korkmaz, 2023). This study demonstrates that the certified adversarial training techniques significantly limit the generalization capabilities of the deep reinforcement learning policies.

## 3 FOUNDATIONS FOR HARMONIC LEARNING

In this section we will introduce harmonic analytic learning and provide the theoretical foundations and analysis for the harmonic learning algorithm. In particular, our algorithm is based on random basis function elimination in a harmonic analytic basis of the state observations during training. Section 4 demonstrates that our theoretically well-founded harmonic learning algorithm results in up to $20\times$ improvement in sample-efficiency. Section 4.1 and 4.3 will further prove that harmonic learning not only improves the sample complexity but further converges to an intrinsically more robust policy. The theoretical analysis for these results lies in the fact that random harmonic analytic basis elimination can be interpreted as a form of value function randomization, a well-established technique with provable guarantees on sample-efficiency in the function approximation setting. Thus, in order to provide a theoretical foundation for harmonic learning, we connect randomized elimination of basis functions to randomized least-squares value iteration (RLSVI), which provides provable regret bounds via randomization of the learned value function. The setting for the provable regret bounds of RLSVI, including many related follow-up studies (Ladosz et al., 2022; Agarwal et al., 2022), is in finite-horizon, episodic MDPs with linear function approximation of the state-action value function. A finite-horizon MDP is represented by $M = (\mathcal{S}, \mathcal{A}, \mathcal{P}, r, H)$ where $\mathcal{S}$ is the set of states, and $\mathcal{A}$ the actions. For each $t \in \{1, \ldots, H\}$, state $s$, and action $a$ the transition function $\mathcal{P}_t(\cdot \mid s, a)$ gives the probability distribution over the next state, and the reward function $r_t(s, a)$ outputs the immediate rewards. Let $\Phi_t : \mathcal{S} \times \mathcal{A} \to \mathbb{R}^\kappa$ represent the feature map such that the state-action value function is given by $\mathcal{Q}_{\theta_t}(s, a) = \Phi_t(s, a)^\top \theta_t$. The RLSVI algorithm proceeds in episodes, where in the $k$-th episode value iteration is performed with a value function that is perturbed by specifically chosen noise $\eta$. In particular, for each episode $i \in \{1, \ldots, k-1\}$ let $(s_{ti}, a_{ti}, r_{ti})$ be the state-action-reward tuple observed at time step $t$. For parameters $\lambda > 0$ and $\sigma > 0$, let $\hat{\theta}_t$ be the parameter estimate for the value function computed via standard least squares value iteration:

$$\hat{\theta}_{t,k} = \arg\min_\theta \left( \frac{1}{\sigma} \sum_{i=1}^{k-1} \left( \Phi_t(s_{ti}, a_{ti})^\top \theta - (r(s_{ti}, a_{ti}) + \max_a \Phi_t(s_{t+1\,i}, a)^\top \theta_{t+1,k})^2 + \lambda \|\theta\|^2 \right) \right)$$

Then define the regularized regression matrix $\Omega_{t,k} = \frac{1}{\sigma^2} \sum_{i=1}^{k-1} \Phi_t(s_{ti}, a_{ti}) \Phi_t(s_{ti}, a_{ti})^\top + \lambda I$. The updated parameters of RLSVI are computed by sampling $\eta_{t,k} \sim \mathcal{N}(0, \Omega_{t,k}^{-1})$ and setting $\theta_{t,k} = \hat{\theta}_{t,k} + \eta_{t,k}$. Intuitively, the Gaussian noise $\eta_{t,k}$ added to the value function parameters is chosen to have larger variance along directions where fewer feature vectors $\Phi_t(s_t, a_t)$ have been observed so far, and lower variance along directions with many previously observed feature vectors. This has

the effect of directly injecting uncertainty into value estimates proportional to a natural posterior distribution on the parameters. In particular, for any state-action pair $(s_t, a_t) \in \mathcal{S} \times \mathcal{A}$ the state-action value under the random perturbation is given by

$$\mathcal{Q}_{\theta_{t,k}}(s_t, a_t) = \Phi_t(s_t, a_t)^\top (\hat{\theta}_{t,k} + \eta_{t,k}) = \Phi_t(s_t, a_t)^\top \hat{\theta}_{t,k} + \Phi_t(s_t, a_t)^\top \eta_{t,k}$$

$\mathcal{Q}_{\theta_{t,k}}(s_t, a_t) = \mathcal{Q}_{\hat{\theta}_{t,k}}(s_t, a_t) + \Phi_t(s_t, a_t)^\top \eta_{t,k}$. Observe that the value $\Phi_t(s_t, a_t)^\top \eta_{t,k}$ has a Gaussian distribution equal to $\mathcal{N}\left(0, \Phi_t(s_t, a_t)^\top \Omega_{t,k}^{-1} \Phi_t(s_t, a_t)\right)$. Therefore, the random perturbation to each state-action value has variance inversely proportional to a measure of the confidence of the current state-action value estimate. In the general function approximation setting, e.g. when using deep neural networks, it is no longer possible to directly compute the correct noise level to perturb the value estimates via an inversion of the feature covariance matrix. However, we will present an alternative approach, i.e. harmonic learning, that transfers more easily to the general function approximation setting, while simultaneously preserving the intuition that the variance should be higher at state-action pairs for which the current $\mathcal{Q}$-function estimate is less confident. To begin we introduce the notion of a stable basis for the feature space of an MDP.

**Definition 3.1** (*Linear Function Approximation Stable Basis*). Let $M$ be a finite horizon MDP with linear function approximation via feature map $\Phi_t : \mathcal{S} \times \mathcal{A} \to \mathbb{R}^\kappa$ and optimal state-action value function $\mathcal{Q}_{\theta_t^*}$. Let $v_1, \ldots, v_\kappa \in \mathbb{R}^\kappa$ be an orthonormal basis and let $\hat{\Phi}_t(s, a)_i = \Phi(s, a)_t^\top v_i$, The set $v_1, \ldots, v_\kappa$ is an $\epsilon$-*stable basis* for $M$ if for all $s \in \mathcal{S}$, $a \in \mathcal{A}$, and $i \in \{1, \ldots, \kappa\}$

$$\left| \frac{1}{\kappa} \Phi_t(s, a)^\top \theta_t^* - \hat{\Phi}_t(s, a)_i v_i^\top \theta_t^* \right| < \epsilon$$

To gain an intuition for Definition 3.1, observe that for any orthonormal basis $v_1, \ldots, v_\kappa$ the feature vector can be written as the linear combination $\sum_i \hat{\Phi}_t(s, a)_i v_i$. Thus, the definition requires that each component $\hat{\Phi}_t(s, a)_i v_i$ of the feature vector along direction $v_i$ contributes approximately a $\frac{1}{\kappa}$ fraction of the optimal state action value $\mathcal{Q}_{\theta_t^*}(s, a) = \Phi_t(s, a)^\top \theta_t^*$. This property is analogous to the uncertainty principle in harmonic analysis, which qualitatively states that signals which are localized with respect to the standard basis must be more evenly spread out with respect to the harmonic analytic basis. Given a stable basis, there is a natural measure of uncertainty for any estimate of the state-action value function.

**Definition 3.2** (*Uncertainty of Parameters*). Let $v_1, \ldots, v_\kappa$ be an $\epsilon$-stable basis for $M$. For any parameter estimate $\theta_t$ for the state-action values the *uncertainty* of $\theta_t$ for action $a$ in state $s$ is

$$\Upsilon_{\theta_t}(s, a) = \frac{1}{\kappa} \sum_i \left( \frac{1}{\kappa} \Phi_t(s, a)^\top \theta_t - \hat{\Phi}_t(s, a)_i v_i^\top \theta_t \right)^2 .$$

Observe that for the optimal parameters $\theta^*$ we have $\Upsilon_{\theta_t^*}(s, a) < \epsilon^2$. In general, the uncertainty measures how far the estimate $\theta_t$ deviates from having an equal contribution from the components of the feature map $\Phi_t(s, a)$ along each of the basis vectors $v_i$. Since the vectors $v_i$ form a stable-basis (and thus the contribution in each component to the optimal state-action value should be equal), larger values for the uncertainty implies that the estimate $\theta_t$ is further from the optimum for action $a$ in state $s$. We now have all the ingredients to introduce the foundations for harmonic learning. Essentially, Algorithm 1 adds noise to the value function by removing the component of the feature $\Phi_t(s, a)$ along a randomly chosen stable-basis direction $v_i$.

---

**Algorithm 1** Stable-basis noise

1: **Input:** A stable basis $v_1, \ldots, v_\kappa$ for an MDP $M$. An estimate $\theta_t$ for the state-action value function, a state $s$, and action $a$.
2: Sample $i$ uniformly at random from $\{1, \ldots, \kappa\}$
3: Set $\tilde{\Phi}_t(s, a) = \Phi_t(s, a) - \hat{\Phi}_t(s, a)_i v_i$
4: Output the noisy state-action value $\tilde{\Phi}_t(s, a)^\top \theta_t$

---

**Proposition 3.3** (*Stable-basis noise variance*). Let $v_1, \ldots, v_\kappa$ be an $\epsilon$-stable basis for $M$. For parameter vector $\theta_t$, let $\tilde{\mathcal{Q}}_{\theta_t}(s, a)$ be the state-action value estimate output by Algorithm 1. Let $\eta = \tilde{\mathcal{Q}}_{\theta_t}(s, a) - \mathcal{Q}_{\theta_t}(s, a)$. Then $\text{Var}[\eta] = \Upsilon_{\theta_t}(s, a)$.

*Proof.* First observe that $\mathbb{E}[\eta] = \mathbb{E}[\tilde{\mathcal{Q}}_{\theta_t}(s,a) - \mathcal{Q}_{\theta_t}(s,a)]$, and

$$\mathbb{E}[\eta] = \mathbb{E}_i\left[(\Phi_t(s,a) - \hat{\Phi}_t(s,a)_i v_i)^\top \theta_t - \Phi_t(s,a)^\top \theta_t\right] = -\mathbb{E}_i\left[\hat{\Phi}_t(s,a)_i v_i^\top \theta_t\right] = -\frac{1}{\kappa}\Phi_t(s,a)^\top \theta_t$$

Therefore, the variance of the noise $\eta$ is given by

$$\mathrm{Var}[\eta] = \mathbb{E}\left[\left(\tilde{\mathcal{Q}}_{\theta_t}(s,a) - \mathcal{Q}_{\theta_t}(s,a) + \frac{1}{\kappa}\Phi_t(s,a)^\top \theta_t\right)^2\right]$$

$$= \mathbb{E}_i\left[\left(\frac{1}{\kappa}\Phi_t(s,a)^\top \theta_t - \hat{\Phi}_t(s,a)_i v_i^\top \theta_t\right)^2\right] = \Upsilon_{\theta_t}(s,a) \qquad \square$$

Random modification of the value function via Algorithm 1 is equivalent to adding noise $\eta$ to $\mathcal{Q}_{\theta_t}(s,a)$ where the variance of the noise $\eta$ is exactly equal to the uncertainty $\Upsilon_{\theta_t}(s,a)$. Thus, by simply deleting the component of the feature vector along a randomly selected stable-basis vector, one can add noise that has variance proportional to a natural uncertainty measure for the current parameter estimate.

### 3.1 GENERAL FUNCTION APPROXIMATION

Now we will extend both the definition of a stable basis and Algorithm 1 to the general function approximation setting. In this setting we will assume that the state space $\mathcal{S}$ is a $d$-dimensional vector space, the action space $\mathcal{A}$ is finite, and the optimal state-action value function $\mathcal{Q}^*(s,a)$ is a general function on $\mathcal{S} \times \mathcal{A}$.

**Definition 3.4** (*General Function Approximation Stable Basis*). Let $M$ be an MDP and $\epsilon > 0$. An $\epsilon$-stable basis for $M$ is an orthonormal basis $v_1, \ldots, v_\kappa$ for $\mathcal{S}$ such that for all $i$,

$$\left|\mathcal{Q}^*(s,a) - \mathcal{Q}^*(s - (v_i^\top s)v_i, a)\right| < \frac{1}{\kappa}\mathcal{Q}^*(s,a) + \epsilon.$$

Algorithm 1 can also be easily modified for the general function approximation setting by sampling a random $i$, and replacing the state $s$ with $\tilde{s} = s - (v_i^\top s)v_i$ i.e. by sampling a random $i$ and deleting the component of $s$ along $v_i$. The following proposition shows that in the general function approximation setting, one can test for the presence of a stable basis by modifying states via Algorithm 1 and measuring cumulative rewards.

**Proposition 3.5** (*Harmonic Stability of the Policy*). Let $v_1, \ldots, v_\kappa$ be an $\epsilon$-stable basis for an MDP $M$. For a state $s$ let $a^*(s) = \arg\max_a \mathcal{Q}^*(s,a)$ be the argmax action. Assume that $\epsilon < \frac{1}{2}\left(\frac{\kappa-1}{\kappa}\mathcal{Q}^*(s,a^*(s)) - \arg\max_{a \neq a^*(s)} \frac{\kappa+1}{\kappa}\mathcal{Q}^*(s,a)\right)$ for all $s \in S$. Let $R^*$ be the expected cumulative rewards when following the argmax policy according to $\mathcal{Q}^*$. Then if each state is modified according to the general function approximation version of Algorithm 1 the expected cumulative discounted rewards $R$ obtained under the argmax policy satisfies $R = R^*$.

*Proof.* Under the general function approximation version of Algorithm 1 each state $s$ encountered is modified to $\tilde{s} = s - (v_i^\top s v_i)$. By Definition 3.4 for the action $a^*(s)$

$$\mathcal{Q}^*(\tilde{s}, a^*(s)) > \mathcal{Q}^*(s, a^*(s)) - \frac{1}{\kappa}\mathcal{Q}^*(s, a^*(s)) - \epsilon = \frac{\kappa - 1}{\kappa}\mathcal{Q}^*(s, a^*(s)) - \epsilon \qquad (1)$$

Similarly, by Definition 3.4, for any $a \neq \arg\max_a \mathcal{Q}(s,a)$

$$\mathcal{Q}^*(\tilde{s}, a) < \frac{\kappa + 1}{\kappa}\mathcal{Q}^*(s,a) + \epsilon \leq \arg\max_{a \neq a^*(s)} \frac{\kappa + 1}{\kappa}\mathcal{Q}^*(s,a) + \epsilon. \qquad (2)$$

Combining (1) and (2) with the assumption on $\epsilon$ implies that $\mathcal{Q}^*(\tilde{s}, a) < \mathcal{Q}^*(\tilde{s}, a^*(s))$ for all $a \neq a^*(s)$. Thus the argmax action in each state under Algorithm 1 is equal to the argmax action in the original unmodified state, implying that the distribution of the trajectory and the cumulative rewards remain unchanged. $\qquad \square$

Table 1: Performance analysis results for BC, GAIL, SQIL, vDICE, harmonic learning and inverse $\mathcal{Q}$-learning. Table reports raw scores obtained by the policies in high-dimensional MDPs.

| MDP | BC | GAIL | SQIL | vDICE | Inverse$\mathcal{Q}$L | Ours |
|---|---|---|---|---|---|---|
| Pong | -11.00 | -20.3±0.008 | -19.81±0.012 | -20.91±0.001 | 8.0±5.3814 | **19.0±1.89736** |
| SpaceInvader | 300 | 289±24.1 | 218±8.9 | 176±12.8 | 470.5±23.68 | **609.0±14.5223** |
| Breakout | 0.00±0.00 | 0.5±0.001 | 3.6±0.021 | 4.2±0.031 | 108.9±29.72 | **228.8± 35.4606** |

The uncertainty principle of Fourier transform states that a function and its Fourier transform can't be sharply concentrated at the same time: if one is very localized, the other must be spread out (Heisenberg, 1927; Benedicks, 1985). Thus, the removal of one element of the Fourier basis is spread out in the function without any semantic changes to the natural image. The optimal robust policy in high-dimensional MDPs should be robust to disturbances that have no true semantic meaning to the human eye, and hence should not have large dependence on any one part of the basis corresponding to a particular frequency.

The results in Section 4 further demonstrate that indeed deep reinforcement learning policies trained with high-dimensional state-observations have this inherent robustness. In particular Figure 2 demonstrates that removing each subset of elements of the basis corresponding to one particular frequency has approximately equal impact on vanilla trained policy performance. On the other hand inverse-$\mathcal{Q}$ learning, which uses fewer environment interactions, has much larger variation in policy sensitivity across frequencies. Thus, the policy that is trained with more interactions, and thus is closer to optimal, verifies the conditions of Definition 3.1 with respect to the harmonic basis. Therefore, we can extend our method from Algorithm 1 to learning from high-dimensional state observations by leveraging randomized removal of the elements of the stable basis, i.e. removing components of the state-observation in the Fourier basis in order to induce uncertainty in the value function to boost sample-efficiency.

---

**Algorithm 2** Harmonic Learning

**Input:** Occupancy measure of the expert policy $\rho_E$, regularizer $\phi$, $\kappa$ dimension of the state observations, $\mu$ experiences from replay buffer, learning rate $\alpha_{\mathcal{Q}}$, actions $a \in \mathcal{A}$, states $s \in \mathcal{S}$, initialize $\mathcal{Q}_{\theta_0}$ and stable basis frequency $\Psi$.

**for** $t = 0$ **to** $N$ **do**
    **for** $s = s_0$ **to** $s_T$ **do**
        Sample $\delta \sim \mathcal{U}(0, \kappa/2)$
        $\mathcal{F}_s(u,v) = \frac{1}{\kappa^2} \sum_{m=0}^{\kappa-1} \sum_{n=0}^{\kappa-1} s(m,n) e^{-j2\pi((um+vn)/d)}$
        $\mathcal{F}_s[\delta, \delta : \kappa - \delta] = \mathcal{F}_s[\kappa - \delta, \delta : \kappa - \delta] = \Psi$
        $\mathcal{F}_s[\delta : \kappa - \delta, \delta] = \mathcal{F}_s[\delta : \kappa - \delta, \kappa - \delta] = \Psi$
        $s^{\text{spc}}(m,n) = \sum_{u=0}^{\kappa-1} \sum_{v=0}^{\kappa-1} \mathcal{F}(u,v) e^{j2\pi((um+vn)/\kappa)}$
        Insert $s^{\text{spc}}$ to the buffer instead of $s$
        Train $\mathcal{Q}$ function:
        $\tilde{\mathcal{V}}(s) = \mathbb{E}_{(s,a)\sim\mu}[\mathcal{V}_{\theta_t}(s)] - \gamma \mathbb{E}_{s'\sim\mathcal{P}(\cdot|s,a)}[\mathcal{V}_{\theta_t}(s')]$
        $\mathcal{Z} = \nabla_{\theta_t}[\mathbb{E}_{\rho_E}\phi(\mathcal{Q}_{\theta_t}(s,a) - \gamma\mathbb{E}_{s'\sim\mathcal{P}(\cdot|s,a)}\mathcal{V}_{\theta_t}(s'))]$
        $\theta_{t+1} \leftarrow \theta_t + \alpha_{\mathcal{Q}}\mathcal{Z} - \alpha_{\mathcal{Q}}\nabla_{\theta_t}\tilde{\mathcal{V}}(s)$
    **end for**
**end for**
**Return:** State-action value function $\mathcal{Q}_{\theta_N}(s,a)$

---

## 4 EXPERIMENTAL ANALYSIS

While Section 3 provides theoretical justification for our proposed training method, in this section we provide details into the harmonic learning algorithm where Algorithm 2 provides pseudocode for harmonic learning. The visualizations of transformations of the elements of the stable basis of state observations are also demonstrated in Figure 3. The experiments provided in our paper are conducted in the Arcade Learning Environment. All of the MDPs considered in our paper have high-dimensional state representations. The deep reinforcement learning policies used in stable basis robustness analysis are trained via double-$\mathcal{Q}$ learning (van Hasselt et al., 2016; van Hasselt, 2010). Natural robustness was measured with the same parameters used in (Korkmaz, 2024). The experiments are conducted with 10 random runs. The standard error of the mean is included in all of the results presented throughout the paper. All of the policies that are trained without the reward signal from the MDP use the exact same hyperparameters with inverse $\mathcal{Q}$-learning algorithm (Section 2) to provide consistent and transparent comparison. See supplementary material for the code, the hyperparameters and architecture details. Table 2 reports the raw scores and human normalized scores for the inverse $\mathcal{Q}$-learning and the harmonic learning algorithm in Pong, Breakout, Seaquest, SpaceInvaders and BeamRider. As Table 2 reports, the performance obtained by harmonic learning over inverse $\mathcal{Q}$-learning in Breakout is 210%. Furthermore, intriguingly harmonic learning can reach

Figure 1: State observations of the high-dimensional state representation MDPs with spectral changes in ALE. The harmonic basis and semantic effects in the MDP.

Table 2: Performance analysis results for harmonic learning and inverse $\mathcal{Q}$-learning in Pong, Breakout, Seaquest, SpaceInvaders and BeamRider. Table reports raw scores and human normalized scores.

| Performance Analysis | Raw Scores | | Human Normalized Scores | |
|---|---|---|---|---|
| Training Method | Harmonic Learning | Inverse $\mathcal{Q}$-learning | Harmonic Learning | Inverse $\mathcal{Q}$-learning |
| Pong | **19.0±1.89736** | 8.0±5.3814 | **1.3233± 0.0199** | 0.9566±0.05672 |
| Seaquest | **906.0±53.2202** | 864.0±42.0285 | **0.04164±0.00083** | 0.03955±0.00066 |
| SpaceInvader | **609.0±14.5223** | 470.555±23.6812 | **0.3064±0.003052** | 0.2144±0.00497 |
| BeamRider | **1023.6±140.974** | 909.6±65.392 | **0.1219±0.0082** | 0.1008±0.0038 |
| Breakout | **228.8± 35.4606** | 108.9±29.7198 | **7.5448±0.37254** | 3.5614±0.3122 |

a score of 19.0 for Pong in **only 50K environment interactions**, where inverse-$\mathcal{Q}$ learning is unable to reach this score even with 1 million environment interactions after convergence. Thus, harmonic learning is not only sample-efficient but further simply converges to a substantially better policy as an end product. Figure 5 reports natural robustness analysis results for harmonic learning and inverse $\mathcal{Q}$-learning. Intriguingly, these results demonstrate that the inverse $\mathcal{Q}$-learning policies results in less generalizable and less robust. Natural robustness results reported in Figure 5 further solidifies that not only does harmonic learning result in sample-efficient learning as demonstrated in Table 2, it further learns more resilient, generalizable and robust policies that can perform well even in non-stationary environments. Furthermore, harmonic learning does not require any gradient or function evaluations and only takes 9 lines of code, hence resulting in an extremely fast and efficient algorithm.

> *Harmonic learning is modular: It is a plug-and-play method with any baseline algorithm.*

### 4.1 STABLE BASIS ROBUSTNESS ANALYSIS OF DEEP SEQUENTIAL DECISION MAKING

Section 4 provides the empirical analysis of the harmonic learning algorithm in high-dimensional complex MDPs, and the results demonstrate that harmonic learning improves sample efficiency by up to $20\times$. Our objective was not only to improve sample complexity but further to construct policies that can make robust and resilient decisions in uncertain non-stationary environments. In this section we will introduce the techniques that quantify the volatilities in decision making. In particular, the objective of Stable Basis Robustness Analysis (SBRA) is to quantify and measure the impact of the elements of the stable basis on the policy performance. Figure 2 demonstrates that deep reinforcement learning policies are stable in harmonic basis satisfying the $\left| \frac{1}{\kappa} \Phi_t(s,a)^\top \theta_t^* - \hat{\Phi}_t(s,a)_i v_i^\top \theta_t^* \right| < \epsilon$.

Upon setting the $\delta$-frequencies to $\Psi$ the discrete Fourier transform is inverted and the observation of the deep neural policy consists of $s^{\mathrm{spc}}$ as in Algorithm 3. For a state $s \in \mathcal{S}$ the discrete Fourier transform of the state $s$ is

$$\mathcal{F}_s(u,v) = \frac{1}{\mathcal{M}\mathcal{N}} \sum_{m=0}^{\mathcal{M}-1} \sum_{n=0}^{\mathcal{N}-1} s(m,n) e^{-j2\pi(um/\mathcal{M}+vn/\mathcal{N})}$$

The impact on the policy performance is measured by $\mathcal{I} = (\mathrm{Score}_{\mathrm{baseline}} - \mathrm{Score}_{\mathcal{F}_s})/(\mathrm{Score}_{\mathrm{baseline}})$, where $\mathrm{Score}_{\mathcal{F}_s}$ represents the score obtained by the policy when the state observations are transformed as described in Algorithm 3, and $\mathrm{Score}_{\mathrm{baseline}}$ represents the score obtained by the baseline policy without any modifications applied to the state observations.

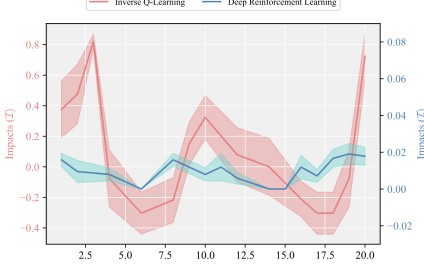

Figure 2: Stable Basis Robustness Analysis (SBRA) results for the deep reinforcement learning and the state-of-the-art deep inverse reinforcement learning.

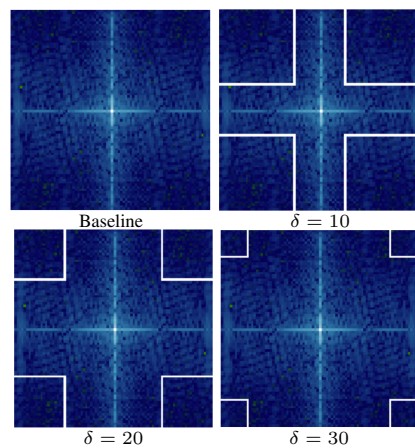

Figure 3: Representations of the Stable Basis Robustness Analysis (SBRA) with variations of $\delta$ on state observations.

**Algorithm 3** SBRA: Stable Basis Robustness Analysis

**Input:** State-action value function $\mathcal{Q}(s, a)$, actions $a \in \mathcal{A}$, states $s \in \mathcal{S}$, stable basis robustness analysis frequency $\Psi$, policy $\pi(s, a)$, $\kappa$ dimension of the state observations
**Output:** Impact on the policy performance
**for** $\delta = 0$ to $\kappa/2$ **do**
    **for** $s = s_0$ to $s_T$ **do**
$$\mathcal{F}_s(u, v) = \frac{1}{\kappa^2} \sum_{m=0}^{\kappa-1} \sum_{n=0}^{\kappa-1} s(m, n) e^{-j2\pi((um+vn)/\kappa)}$$
$$\mathcal{F}_s[\delta, \delta : \kappa - \delta] = \mathcal{F}_s[\kappa - \delta, \delta : \kappa - \delta] = \Psi$$
$$\mathcal{F}_s[\delta : \kappa - \delta, \delta] = \mathcal{F}_s[\delta : \kappa - \delta, \kappa - \delta] = \Psi$$
$$s^{\text{spc}}(m, n) = \sum_{u=0}^{\kappa-1} \sum_{v=0}^{\kappa-1} \mathcal{F}(u, v) e^{j2\pi((um+vn)/\kappa)}$$
$$\hat{a}(s) = \arg\max_{a \in \mathcal{A}} \mathcal{Q}(s^{\text{spc}}, a)$$
    **end for**
**end for**
**Return:** Impact $\mathcal{I}$

Figure 2 reports results on the stable basis robustness analysis of the deep reinforcement learning policy and the deep inverse reinforcement learning policy as the randomized $\delta$-frequencies are transformed to $\Psi$. Figure 3 provides the steps of SBRA with variations of $\delta$. The results reported in Figure 2 demonstrate that vanilla trained deep reinforcement learning policies are more robust than the policies trained via deep inverse reinforcement learning. In particular, there is a high increase in the sensitivities towards lower frequencies for the deep inverse reinforcement learning policy.

## 4.2 Overfitting of State-Action Value Function in Inverse Reinforcement Learning

The results reported in this section demonstrate that inverse $\mathcal{Q}$-learning assigns higher state-action values than harmonic learning, even though the rewards obtained are lower compared to harmonic learning. In particular, Figure 4 reports the state-action value of the action maximizing the $\mathcal{Q}$-function in a given state for deep neural policies trained via harmonic learning and inverse $\mathcal{Q}$-learning. Table 3 reports the average total rewards obtained and the average state-action values of the actions that maximize the state-action value function in a given state (i.e. $\mathbb{E}_{s \sim e(s), e \sim \varepsilon(e)}[\max_a \mathcal{Q}(s, a)]$) for harmonic learning and inverse $\mathcal{Q}$-learning policies. The fact that inverse $\mathcal{Q}$-learning policies construct a state-action value function that assigns higher values while the true rewards obtained are lower demonstrates that inverse $\mathcal{Q}$-learning results in learning overestimated state-action values. These results demonstrate that harmonic learning further targets the overfitting problem of the state-action value function and results in lowering the overestimation bias in state-action values compared to the baseline training methods.

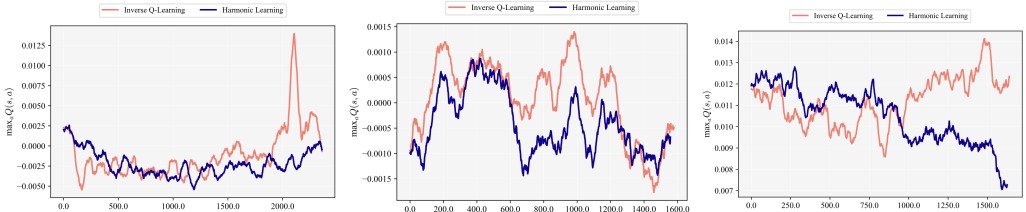

Figure 4: State-action values for the deep neural policies trained via harmonic and inverse $\mathcal{Q}$-learning for BeamRider, Pong and Breakout.

## 4.3 Natural Robustness and Generalization

In this section we provide a detailed analysis on the robustness of deep sequential decision making policies to distributional shift. In particular, recent work connected the relationship between adversarial robustness and natural robustness in a given MDP in terms of the damage caused by these natural directions in the deep neural policy landscape on the policy performance and the perceptual similarity distances to the base state observations (Korkmaz, 2023). In particular, the imperceptibility $\mathcal{P}_{\text{similarity}}$

Table 3: Average rewards obtained and average state-action values of the actions maximizing the state-action value function in a given state (i.e. $\mathbb{E}_{s\sim e(s),e\sim\varepsilon(e)}[\max_a \mathcal{Q}(s,a)]$) for harmonic learning and inverse $\mathcal{Q}$-learning policies in Pong, Breakout, SpaceInvaders and BeamRider.

| $\mathcal{Q}$ Analysis | $\mathbb{E}_{s\sim e(s),e\sim\varepsilon(e)}[\max_a \mathcal{Q}(s,a)]$ | | Average Rewards | |
| --- | --- | --- | --- | --- |
| Method | Harmonic Learning | Inverse $\mathcal{Q}$-learning | Harmonic Learning | Inverse $\mathcal{Q}$-learning |
| SpaceInvader | **0.001291±0.0001532** | -0.000188±7.55×10$^{-5}$ | **602.0±13.023056** | 528.5±18.9347 |
| BeamRider | **-0.001739±4.34×10$^{-5}$** | -0.001808±2.17×10$^{-5}$ | **1108.4± 158.10725** | 908.8±95.039865 |
| Breakout | 0.009761±6.25×10$^{-5}$ | **0.01085±3.70×10$^{-5}$** | **214.3±38.5888** | 39.0±6.1967 |
| Pong | -0.0007503±3.67×10$^{-5}$ | **-0.000455±7.60×10$^{-5}$** | **19.0±1.89736** | 8.0±5.3814 |

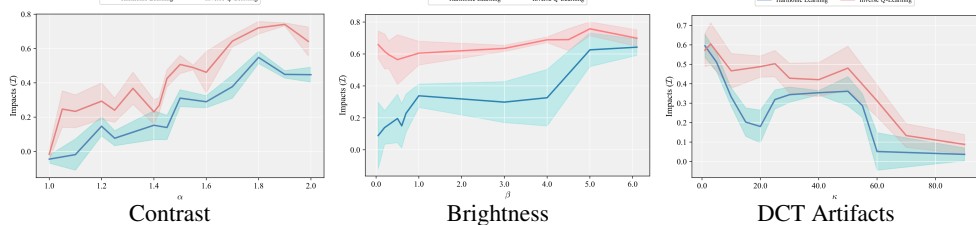

| Contrast | Brightness | DCT Artifacts |

Figure 5: Impact values for natural robustness analysis with non-robust directions intrinsic to the MDP for discrete cosine transform artifacts, brightness and contrast reporting generalization and robustness results for harmonic learning and inverse $\mathcal{Q}$-learning policies in high dimensional MDPs.

is measured by, $\mathcal{P}_{\text{similarity}}(s, s + \xi(s,\pi)) = \sum_l \frac{1}{H_l W_l} \sum_{h,w} \|w_l \odot (\hat{y}^l_{shw} - \hat{y}^l_{(s+\xi(s,\pi))hw})\|_2^2$ where $\hat{y}^l_s, \hat{y}^l_{\hat{s}} \in \mathbb{R}^{W_l \times H_l \times C_l}$ represent the vector of activations in the convolutional layers with width $W_l$, height $H_l$, $C_l$ is the number of channels, and $\xi(s,\pi)$ natural change function Zhang et al. (2018)[1]. Note that Section 4.1 demonstrates in detail that policies that learn from observing an expert without having access to the reward function are less robust compared to deep reinforcement learning policies. In many prominent settings, e.g. large language models as in RLHF and self driving cars, constructing a reward function is substantially more difficult than learning one, and achieving more sample efficient and robust policies carries significant importance given both the safety and security concerns that have been recently raised. Hence, in this section we provide a comprehensive investigation on the robustness and Figure 5 reports the performance profile results under these natural directions. In particular, the results reported in Figure 5 demonstrate that harmonic learning results in intrinsically more robust deep neural policies that can generalize to observations that have not been seen before by the policy (i.e. points that are outside of the training environment). These results once more demonstrate that harmonic learning not only results in learning sample-efficient policies but also further learns policies that are both robust and generalizable.

## 5 CONCLUSION

In this paper we aim to seek answers for the following questions: *(i) How can we build sample efficient deep neural policies that learn via observing experts as robust as a policy that learns via exploration in high-dimensional MDPs? (ii) Is it possible to simultaneously improve sample efficiency without sacrificing robustness?* To be able to address these questions, we propose a theoretically well-founded training algorithm that leverages the spectral analysis and perspective in deep sequential decision making. We conduct extensive experiments in the Arcade Learning Environment and the empirical analysis demonstrates that our proposed algorithm results in exceptional sample efficiency improvement. Moreover, we propose a novel method that provides a comprehensive analysis of the robustness and instabilities of deep neural policies. We provide further extensive investigation on the state-action value function learned by the deep neural policies and demonstrate that prior methods suffer from overestimation problems. Furthermore, we conduct a robustness analysis that investigates the response of the deep sequential decision making policies to distributional shift. The results provided in our paper demonstrate that the theoretically well-founded harmonic learning algorithm we introduce results in exceptionally sample-efficient, and furthermore substantially robust and resilient deep neural policies.

---

[1]In connection with the contradistinction between adversarial and natural robustness, recently it has been shown that standard reinforcement learning policies are more robust and can generalize better compared to certified robust (i.e. adversarial) trained ones by the natural robustness framework (Korkmaz, 2023).

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
