# OpenReview forum: "Stable Basis Deep Neural Policy Training"
_ICLR.cc/2026/Conference — Submitted to ICLR 2026_

### Official Review · Reviewer_fqjL · 2025-10-31

**Soundness:** 2
**Presentation:** 2
**Contribution:** 2
**Rating:** 2
**Confidence:** 2

**Summary:**

The paper introduces a new reinforcement learning framework called Harmonic Learning, which enables sample-efficient and robust policy training in high-dimensional Markov Decision Processes (MDPs) . Training occurs without access to explicit reward signals. The key contribution is in defining a stable harmonic basis of the state representation, and introducing a noise-injection procedure that improves both learning efficiency and policy robustness.

**Strengths:**

1. Interesting spectral perspective on policy learning: The paper introduces a novel spectral viewpoint for policy training by leveraging harmonic or Fourier basis perturbations. While the theoretical underpinnings are partly heuristic, the idea of connecting uncertainty quantification in reinforcement learning to harmonic analysis is conceptually appealing and provides a fresh way to study robustness in high-dimensional MDPs.

2. Empirical consistency and sample-efficiency improvement: The experiments, conducted across several Atari environments demonstrate that the proposed method can achieve comparable or superior performance with significantly fewer interactions (reported as up to 20× sample-efficiency gains in Tables 1–2 and §4 of the main paper.

**Weaknesses:**

1. Presentation is lacking. Currently when I read the paper, in first reading it is not clear to me what exactly is being addressed in this work: is it training policies in absence of reward signals? Is it learning robust policies? or is it a reduced sample complexity? Several details are lacking, I have listed these in Questions section.

2. No discussion, or intutition is provided as to why the algorithm is supposed to work. Also, no comparison is provided with previous existing works. I was expecting some "sample complexity type result" which would quantify the improvements, but this is lacking.

3. Theoretical framework lacks formal convergence guarantees: While the paper connects harmonic learning to randomized least-squares value iteration (RLSVI), this connection is conceptual rather than rigorous. No formal convergence theorem or regret bound is provided for the proposed algorithm beyond analogy to RLSVI. Even these derivations are relatively straightforward, and not surprising.

4. Lack of formal theoretical link between harmonic perturbations and robustness (Main paper 3.1–4.1, lines 270–323 & 370–420; Supplement 6, lines 260–310): While the paper claims that removing Fourier components during training induces robustness through the uncertainty principle, this connection remains conceptual and empirically motivated rather than theoretically substantiated. In the main paper (Sections 3.1 and 4.1), robustness is justified by analogy to harmonic analysis. Loose argument is made that a policy insensitive to frequency-specific perturbations is inherently robust. However, no formal analysis is given connecting the removal of spectral components to quantifiable robustness metrics such as Lipschitz smoothness, stability under distributional shift, or bounded change in Q-values. Evidence relies entirely on empirical visualizations (Figures 2, 3) and heuristic reasoning. The supplementary material ( Section 6, "Theoretical Basis and Empirical Analysis", lines 260-310) reiterates the same argument; that "removal of one element of the Fourier basis is spread out in the function without any semantic changes to the natural image". It then infers that this property implies robustness. But the supplement still provides no formal robustness guarantee, bound, or theorem linking harmonic perturbations to policy resilience. The discussion equates empirical stability of scores under spectral deletions (SBRA) with theoretical robustness, without proving this correspondence.

**Questions:**

1. line 92: should it be \theta_t in r.h.s.?

2. "Despite recent progress, a fundamental assumption persists in reinforcement learning: that the agent has
direct access to the reward function of the MDP."--> in RL assumption typically is knowledge of reward obtained i.e. r(s_t,a_t), and not the knowledge of the reward function itself r: S \times A \mapsto R.

3. Please explain this "Orthogonal to these advances while the instabilities of deep neural networks under non-robust directions have been
a subject of discussion (Goodfellow et al., 2015), recent work demonstrated that these instabilities are currently also present in deep neural policies (Huang et al., 2017)." What is meant by "instability" here? Also this "Furthermore, more recent studies demonstrated that these non-robust directions can be semantically meaningful changes to the environment (Korkmaz, 2024)."--> Its hard to understand for someone not working in this exact same field.

4. "robust decisions in unstable and non-robust environments?" whats an unstable environment?

5.  "Harmonic learning achieves substantial sample-efficiency resulting in requiring up to 20× fewer samples while achieving better performance"--> is this 20x improvement theoretically shown or empirical?

6. "spectrally analyze the robustness of deep neural policies"--> more details can be provided (which theorem proves this? or what does this statement mean).

7. "to as an imitation and inverse reinforcement learning algorithm interchangeably" --> why so, how is imitation related to inverse RL?

---

> ### Author Response · Authors · 2025-11-25
>
> We are truly glad to hear you noting that our paper introduces a novel conceptually appealing spectral viewpoint and provides a fresh way to study robustness in high-dimensional MDPs with empirical consistency and significant sample-efficiency improvement that achieves superior performance with significantly fewer interactions. We appreciate you highlighting the clear contributions of our paper.
>
> 1. *”Please explain this "Orthogonal to these advances while the instabilities of deep neural networks under non-robust directions have been a subject of discussion (Goodfellow et al., 2015)... Its hard to understand for someone not working in this exact same field.*”
>
> This part describes prior work that showed vulnerabilities of reinforcement learning policies. These vulnerabilities range from worst-case perturbations to semantically meaningful and naturally plausible perturbations.
>
> 2. *“No discussion, or intutition is provided as to why the algorithm is supposed to work. Also, no comparison is provided with previous existing works.”*
>
> We would like to kindly point out that the results reported in Table 1 show comparisons to a large portfolio of prior work and we have provided intuition and insights on why the algorithm works throughout Section 3 with theoretical analysis, justification and motivation between Line 184-192, Line 157-177, Line 197-204.
>
> 3. *”Theoretical framework lacks formal convergence guarantees: While the paper connects harmonic learning to randomized least-squares value iteration (RLSVI), this connection is conceptual rather than rigorous. No formal convergence theorem or regret bound is provided for the proposed algorithm beyond analogy to RLSVI.”*
>
> We would like to kindly point out that the objective of our paper is not to prove regret bounds on algorithms for the linear function approximation setting, but rather to design effective deep reinforcement learning algorithms with rigorous foundations whose theoretical underpinnings offer concrete, practical guidelines and mechanistic understanding that can directly inform and justify deep reinforcement learning algorithm design. RLSVI applies only to the linear function approximation setting, while our goal is to design an algorithm that works well in deep reinforcement learning. Our results in deep reinforcement learning demonstrate that we indeed achieve this goal, and our theoretical results provide the concrete underpinning on why our algorithm works.
>
>
> 4. *"robust decisions in unstable and non-robust environments?" whats an unstable environment?*
>
> The environment that our agent interacts with after training might not be the exact same environment that the agent is trained in and can contain uncertainty and non-robustness. As also we explained just right above this could be in the form ranging from worst-case changes to distributional shift.
>
> 5. *”Lack of formal theoretical link between harmonic perturbations and robustness”*
>
> We have tested the robustness of the policy with recent natural black-box adversarial attacks that can pierce through known defences, i.e. certified robust training, in deep reinforcement learning and we have reported these results in Figure 5. These results also represent the stability under distributional shift due to the nature of these adversarial attacks. Please also further note that, theoretical guaranteed robustness as in [5] is known to have critical limitations in practice as has been demonstrated by several recent works revealing vulnerabilities of these models. Hence, it is more than ever critical to demonstrate true robustness to accurately and comprehensively assess a model's utility and resilience since theoretical guarantees of robustness currently have limited ability to capture actual robustness of the models [1,2,3].
>
> [1] Increasing Confidence in Adversarial Robustness Evaluations, NeurIPS 2022.
>
> [2] On Adaptive Attacks to Adversarial Example Defenses, NeurIPS 2020.
>
> [3] Obfuscated Gradients Give a False Sense of Security: Circumventing Defenses to Adversarial Examples, ICML 2018.
>
> [4] Adversarial Robust Deep Reinforcement Learning Requires Redefining Robustness, AAAI 2023.
>
> [5] Robust Deep Reinforcement Learning against Adversarial Perturbations on State Observations, NeurIPS 2021.

---

### Official Review · Reviewer_GMNU · 2025-10-31

**Soundness:** 1
**Presentation:** 2
**Contribution:** 3
**Rating:** 2
**Confidence:** 3

**Summary:**

Observations in Reinforcement Learning (RL) environments such as the Arcade Learning Environment (ALE, Bellemare et al. 2013) are often image-based. This paper argues that a robust policy should not depend strongly on non-perceptual features of the observation. In particular, that the policy should be robust to the removal of any particular frequency component in Fourier space.

In the linear function approximation setting, they connect the removal of random basis vectors to Randomised Least Squares Value Iteration [(RLSVI; Osband et al. 2016)](https://arxiv.org/abs/1402.0635), and generalise this to the general function approximation setting by randomly masking certain Fourier components of each observation, before learning the value function using a standard RL algorithm. The resulting algorithm is dubbed Harmonic Learning.

In experiments on the ALE, Harmonic Learning improves on the baseline IQ-Learn [(Garg et al. 2021)](https://arxiv.org/abs/2106.12142) for inverse RL, resulting in improved sample efficiency (up to 50K env steps).

Finally, they propose Stable Basis Robustness Analysis (SBRA), an analysis of the policy performance robustness when perturbing the Fourier components of the observation, as in Harmonic Learning.

**Strengths:**

The viewpoint of this paper is rather original. The impact of image-based perturbations of the observations in Deep RL is a relatively understudied area, and this paper connects the random removal of basis vectors (in the linear function approximation setting) to Randomised Least Squares Value Iteration (RLSVI). The connection to RLSVI is interesting (however please see the related question).

**Weaknesses:**

**1) Positioning within existing literature**

The motivation and setting in the introduction — the utility of RL methods which do not have direct access to the reward –– appears to be mostly unrelated to the main work apart from the fact that the empirical results are built on top of an inverse RL algorithm.

The work would be better positioned directly in the context of robustness to perturbations of the image observations. In this context, there are several related works which consider the impact of perturbations of image inputs in the Fourier space on RL, in particular [(Huang et al. 2022)](https://proceedings.neurips.cc/paper_files/paper/2022/file/802a4350ca4fced76b13b8b320af1543-Paper-Conference.pdf), which introduces "Spectrum Random Masking", which also modifies the observations by masking out components in the Fourier domain. Also see [(Lee et al. 2025)](https://ieeexplore.ieee.org/document/10833629).


**2) Connection to RLSVI**

Please see the question in the later section about the connection to RLSVI in the general function approximation setting.

**3) Empirical evaluation**

The comparison with Inverse Q-Learning are only reported at 50K environment interactions, while previous benchmarks report performance up to 1M. With the exception of Pong, the performance of Harmonic Learning is significantly lower than the final performance achieved by IQ-Learn (at 1M timesteps).

Therefore, it is unclear if Harmonic Learning continues to learn and match (or exceed) that of e.g. IQ-Learn. While it is not necessary to outperform the state-of-the-art in every metric, understanding the limitations of the proposed method would be informative.

**4) Stable Basis Robustness Analysis experiments**

The experiments conducted in Sec 4.1-4.2 propose the use of SBRA as a general diagnosis tool for overfitting and robustness in RL. However, the experiments compare a single baseline algorithm and its Harmonic Learning counterpart. To establish a correlation that generalises beyond this specific pair of algorithms, it would be necessary to compare several inverse RL algorithms and their Harmonic Learning counterpart.

In addition, for the state-action values experiment (Fig 4), it is difficult to draw conclusions by comparing the Q-value estimates, without access to the true Q-values at those states. The true Q-values can be estimated via Monte Carlo rollouts (see e.g. [(Chen et al. 2021)](https://arxiv.org/abs/2101.05982) and references within).

**Questions:**

1. Is seems that it is possible to apply Harmonic Learning to more general standard RL setup, beyond Inverse RL. Did the authors also try this, or is there some reason that Harmonic Learning cannot be applied to the typical RL setup?

2. RLSVI [(Osband et al. 2016)](https://arxiv.org/abs/1402.0635) samples over a posterior distribution over value functions. In contrast, Harmonic Learning samples from a distribution of states. Proposition 3.3 makes the connection with RLSVI explicit for the linear function approximation setting but there does not appear to be an analogous proposition for the general function approximation setting. It would be appreciated if the authors could clarify this.

---

> ### Author Response · Authors · 2025-11-25
>
> We are glad to hear you stating that the viewpoint of our paper is original with good contributions while providing an interesting analysis by connecting the random removal of basis vectors to Randomised Least Squares Value Iteration and our proposed algorithm achieves improved sample efficiency.
>
> 1. *”The work would be better positioned directly in the context of robustness to perturbations of the image observations. In this context, there are several related works which consider the impact of perturbations of image inputs in the Fourier space on RL, in particular (Huang et al. 2022), which introduces "Spectrum Random Masking", which also modifies the observations by masking out components in the Fourier domain. Also see (Lee et al. 2025).*”
>
> We would like to kindly point out that regarding our algorithm and Fourier components, the random removal of the Fourier component in our algorithm has a direct impact on what state the policy transitions to, what action the policy takes and what reward the policy receives. The actions selected and the rewards received by the policy are directly affected by the spectral transformation.
>
> In contrast the study [1] solely applies data augmentation methods in the spectral domain just to increase the amount of data without affecting actions, transitions and rewards. Thus, while [1] is essentially a different variant of data augmentation, on the contrary our algorithm drives more efficient learning and exploration with theoretically solid analysis where a key component of our theoretical analysis in Section 3 is based on the fact that stable-basis noise added to the value function must affect action selection and hence the transitions and the rewards in order to enhance efficiency.
>
> [1] Yangru Huang, Peixi Peng, Yifan Zhao, Guangyao Chen, Yonghong Tian. Spectrum Random Masking for Generalization in Image-based Reinforcement Learning, NeurIPS 2022.
>
> 2. *”Is seems that it is possible to apply Harmonic Learning to more general standard RL setup, beyond Inverse RL. Did the authors also try this, or is there some reason that Harmonic Learning cannot be applied to the typical RL setup?*”
>
> Harmonic learning is a modular algorithm and can be used with any baseline reinforcement learning algorithm. The reason in this paper we focus on inverse RL is due to the critical robustness difference between inverse RL and reinforcement learning.
>
> 3. *”RLSVI (Osband et al. 2016) samples over a posterior distribution over value functions. In contrast, Harmonic Learning samples from a distribution of states. Proposition 3.3 makes the connection with RLSVI explicit for the linear function approximation setting but there does not appear to be an analogous proposition for the general function approximation setting. It would be appreciated if the authors could clarify this.*”
>
> Note that theoretical foundations of RLSVI solely apply to the linear function approximation setting, and hence the formal mathematical analysis given by Proposition 3.3 can only be made in this setting. Nonetheless, one can take the analogy that RLSVI samples a posterior over value functions, and then consider whether Harmonic Learning can be interpreted in this way. In fact, our results for non-linear function approximation (Definition 3.4 and Proposition 3.5) suggest that it can. Hence, these results show that we can view adding stable basis noise as a posterior over the value function which produces a wider distribution of values where insufficient training has occurred, but a very concentrated distribution when the value function is already accurate. This is precisely what the posterior distribution in RLSVI does in the linear case.

---

### Official Review · Reviewer_ybKU · 2025-11-01

**Soundness:** 4
**Presentation:** 2
**Contribution:** 3
**Rating:** 8
**Confidence:** 2

**Summary:**

This paper introduces *harmonic learning* in the context of learning policies in reinforcement learning problems without access to a reward signal, analogous to the inverse RL or imitation learning literature. Using spectral analysis, the authors propose a theoretically sound method for robust Q-learning that can be readily used on top of inverse Q-learning, a current state-of-the-art method for inverse RL/imitation learning, to provide a more sample efficient and robust algorithm for learning RL policies without reward signals.

**Strengths:**

- The authors propose a novel add-on applied to inverse Q-learning that stabilizes training, and greatly improves the sample efficiency of existing methods.
- Their method is well founded theoretically, and the authors provide empirical results that both validate their method based on performance, and analyze how and why harmonic learning improves performance.

**Weaknesses:**

- The experiments are limited to the a handful of games in the Arcade learning environment. It could be nice to have additional experiments in robotics physics based simulators such as Mujoco to better understand whether the improvements of harmonic learning also extend to low-dimensional observation spaces as well.
-  The use of figures and tables in the main text is sometimes not very efficient and unclear as to how it relates to the main text. For example, it is unclear what conclusion is supposed to be drawn from Figures 1 and 3. Table 1 is not referenced in the main text. The final performances achieved from their harmonic learning method is repeated in Table 1, Table 2 and Table 3.

**Questions:**

- Since harmonic learning is modular, can it be used for any Q-learning algorithm? Why is this work limited to the inverse RL problem? If an advantage of the method is that it is modular, could you also apply the method on top of other methods such as BC, GAIL, SQIL, and vDice and see similar improvements?
- One of the main claims of the work is that harmonic learning is much more sample efficient. Claims about sample efficiency can be visualized through learning curves, it would be nice for the authors to include some to further illustrate this point
- The behavior cloning results in Table 1 show no standard errors, except for the breakout environment, where the result is 0. +/- 0. Did something happen with the experiments for BC? Did it fail completely? If so, why?
- Do the results in Figure 4 report a single seed? If so,  it is not clear from a single seed whether claims about value overestimation can be made from this plot alone. If this plot shows an aggregation over multiple seeds, it would be beneficial to add some statistics about the variance as a shaded area.

---

> ### Author Response · Authors · 2025-11-25
>
> We are truly glad to hear that you found our method novel and theoretically sound that stabilizes training and greatly improves the sample efficiency of existing methods while providing empirical results that both validate our method based on performance, and analyze how and why harmonic learning improves performance. We highly appreciate you valuing the soundness of our work as excellent and thank you very much for your insightful and in depth review.
>
> 1. *”Since harmonic learning is modular, can it be used for any Q-learning algorithm? Why is this work limited to the inverse RL problem? If an advantage of the method is that it is modular, could you also apply the method on top of other methods such as BC, GAIL, SQIL, and vDice and see similar improvements?”*
>
> Harmonic learning is a modular algorithm and can be used with any baseline reinforcement learning algorithm. It can indeed also be applied on top of prior methods. Since inverse $Q$-learning was already the state-of-the-art, to provide a consistent and fair comparison we have provided results with the state-of-the-art.
>
> 2. *”The behavior cloning results in Table 1 show no standard errors, except for the breakout environment, where the result is 0. +/- 0. Did something happen with the experiments for BC? Did it fail completely? If so, why?”*
>
> BC results for Breakout can also independently be verified in Figure 4 of [1]. These results also demonstrate that BC is unable to learn a policy in Breakout.
>
> [1] IQ-Learn: Inverse soft-Q Learning for Imitation, NeurIPS 2021. [Spotlight Presentation]
>
> 3. *”Do the results in Figure 4 report a single seed? If so, it is not clear from a single seed whether claims about value overestimation can be made from this plot alone. If this plot shows an aggregation over multiple seeds, it would be beneficial to add some statistics about the variance as a shaded area.”*
>
> Table 3 reports the information provided in Figure 4 with multiple seeds. We thought it would be better for visualization to report Figure 4 as it is. But we can also report as in Table 3.
>
> Thank you very much again for your in depth review and your constructive comments. We would be more than happy to incorporate your suggestions.

---

### Comment · Area_Chair_FMDX · 2025-11-27
**Reminder:  Please Discuss**

All Reviewers,

Thank you for your time. Please engage in discussions with the authors and with one another. There are only a few days left before December 3.

Best,
Area Chair

---

### Meta-Review · Area_Chair_T8Yx · 2025-12-24

**Summary:**

This work considers the deep RL problems where rewards are unavailable or too expensive to compute and proposes a harmonic learning framework for this setting. The authors claim that the resulting algorithm offers theoretical guarantees and substantially improves sample complexity, stability, and generalization in high-dimensional environments.

The reviewers raised several concerns, part of which has been clearly addressed by the authors. However, there are still a few significant points that the authors have not clarified. For example, the paper claims to provide theoretically sound algorithms for deep RL, but the paper does not really extend the theory of RLSVI to the deep RL. Theoretical connections are only done for linear approximation, while everything on neural approximation still remains informal analogy. Similarly, the paper claims improving sample complexity, which has quite formal definitions in theoretical RL, yet the paper only informally refer to practical sample efficiency in the tested experiments. When the reviewer asks for theoretical link between harmonic perturbation and robustness, the authors argue that "theoretical guarantees of robustness currently have limited ability to capture actual robustness of the models" and hence practical evidence is more important. This again, to some degree, contradict the authors emphasis on providing theoretically solid algorithms. The authors may need to rethink how they present and what they claim.

The paper has meaningful contribution, yet it still requires substantial improvement to be published. Therefore, the AC suggests rejecting the paper.

**Reviewer Concerns:**

Addressed:
(1) GMNU: Positioning within existing literature.


Not Addressed:
(1) ybKU & GMNU: Lack of sufficient experimental validation.
(2) GMNU: Lack of compared benchmarks (comment not responded).
(3) fqjL: Several writing and presentation issues.
(4) fgjL: Lack of formal theoretical link between harmonic perturbations and robustness.
(5) fgjL: Lack of formal theoretical convergence guarantees.
(6) GMNU & fgjL: Connection to RLSVI is not formally established.

**Reviewer Scores:**

For all the reviewers, they either do not have much concern, or their concerns are only partially answered. Therefore, the AC does not expect much change in the score. Therefore, the overall score of the paper is still below the conference's threshold.

---

### Decision · Program_Chairs · 2026-01-26

Reject